# Genome-Wide Association Analysis Uncovers Genes Associated with Resistance to Head Smut Pathotype 5 in Senegalese Sorghum Accessions

**DOI:** 10.3390/plants13070977

**Published:** 2024-03-29

**Authors:** Ezekiel Ahn, Louis K. Prom, Sunchung Park, Zhenbin Hu, Clint W. Magill

**Affiliations:** 1USDA-ARS Sustainable Perennial Crops Laboratory, Beltsville Agricultural Research Center, Beltsville, MD 20705, USA; sunchung.park@usda.gov; 2USDA-ARS Southern Plains Agricultural Research Center, College Station, TX 77845, USA; louis.prom@usda.gov; 3USDA-ARS Animal Genomics and Improvement Laboratory, Beltsville Agricultural Research Center, Beltsville, MD 20705, USA; zhenbin.hu@usda.gov; 4Department of Plant Pathology & Microbiology, Texas A&M University, College Station, TX 77843, USA

**Keywords:** sorghum, Senegalese sorghum, head smut, *Sporisorium reilianum*, GWAS, pathotype 5

## Abstract

A newly documented pathotype 5 of the soil-borne fungus *Sporisorium reilianum*, causing head smut in sorghum, was tested against 153 unexplored Senegalese sorghum accessions. Among the 153 sorghum accessions tested, 63 (41%) exhibited complete resistance, showing no signs of infection by the fungus. The remaining 90 accessions (59%) displayed varying degrees of susceptibility. Sorghum responses against *S. reilianum* were explored to analyze the potential link with previously known seed morphology-related traits and new phenotype data from 59 lines for seed weight. A genome-wide association study (GWAS) screened 297,876 SNPs and identified highly significant associations (*p* < 1 × 10^−5^) with head smut resistance in sorghum. By mapping these significant SNPs to the reference genome, this study revealed 35 novel candidate defense genes potentially involved in disease resistance.

## 1. Introduction

Sorghum (*Sorghum bicolor* (L.) Moench) is emerging as a model crop for functional genetics and genomics of tropical grasses with abundant uses, including food, feed, and fuel [1]. Despite its status as a vital cereal crop for millions of people living in about 30 countries in Africa and Asia’s subtropical and semi-arid regions [2], sorghum production faces limitations, especially under climate change pressures. Though often regarded as a substitute for rice and wheat, this adaptable crop (adapting to diverse land conditions) still struggles with various biotic and abiotic stresses, hindering its potential to address the looming global food crisis [2,3,4].

Head smut, a significant disease of sorghum globally, is caused by the soil-borne, facultative biotrophic basidiomycete *Sporisorium reilianum* (Kühn) Langdon and Fullerton (synonyms: *Sphacelotheca reiliana* (Kühn) and G.P. Clinton and *Sorosporium reilianum* (Kühn) McAlpine) [5,6]. Head smut has been identified as a prominent disease affecting sorghum production in Ethiopia, along with anthracnose [7]. The head smut pathogen typically infects the plants during the seedling stage, but symptoms are not expressed until panicle initiation [8]. Unlike most smut pathogens, *S. reilianum* can evade control through conventional seed treatment methods, primarily due to the highly resilient dormancy of its teliospores [9,10]. This unique challenge makes host resistance the sole practical option for managing this disease [10]. While existing resistant sources from the US and Asia offer promising leads for future head smut control strategies [8,11], the ever-evolving nature of pathogens necessitates the continual exploration of new resistance genes. In 2011, two novel *S. reilianum* pathotypes (P5 and P6) emerged amongst isolates collected from South Texas, a region known for its intensive sorghum grain production [5]. The emergence of new pathotypes P5 and P6 and other factors, such as unpredictable weather changes, requires deploying host resistance and identifying new sources from the unexplored diversified sorghum germplasm [8].

Sorghum’s susceptibility to fungal diseases extends beyond direct yield losses, often intertwining with other agronomically important traits. For instance, head smut incidence positively correlates with seed weight [12], suggesting that heavier seeds might be more susceptible. Similarly, grain mold showed strong associations with kernel hardness and pericarp color [13]. Studies have also observed negative correlations between anthracnose severity and both days to anthesis [14] and plant height [15]. Mold-resistant cultivars exhibited significantly greater hardness than mold-susceptible cultivars [16].

Craig and Frederiksen [17] developed a fast and effective way to inoculate sorghum seedlings with head smut. They infected sorghum plants grown in peat pellets with teliospore cultures around their epicotyls and then placed them in vermiculite. The seedlings were submerged in test tubes four days later, allowing the first leaf to develop symptoms. This helped distinguish between susceptible and resistant genotypes (brown/dark spots). Ahn et al. [18,19] used this method to screen Senegalese sorghum accessions for resistance against P5 head smut strains, analyzing both the rate and timing of spot appearance.

Genome-wide association studies, a test involving genetic variants across the genomes of many individuals to identify genotype–phenotype associations [20], have emerged as powerful tools for identifying candidate genes associated with complex traits, including disease resistance [6,21]. Girma et al. [21] employed GWAS in an Ethiopian sorghum landrace collection to identify top candidate defense-related genes against head smut. These included genes encoding glycosyltransferase 3, an MYB family transcription factor, a MADS-box protein, autophagy-related protein 8C precursor, RWP-RK domain, aminomethyl transferase, lipoate-protein ligase B containing protein, and a transcription termination factor 2 [21]. Similarly, Ahn et al. [6] used GWAS with a sorghum mini-core collection to identify candidate defense genes against head smut inoculation. Their findings included genes encoding a leucine-rich repeat, minor histocompatibility antigen H13, protein tyrosine kinase, glycosyl transferase, tetratricopeptide repeat, xyloglucan fucosyl transferase, glutathione S-transferase, aspartyl proteases and xylanase inhibitor N-terminal containing, and a zinc finger containing protein [6].

West and Central African sorghum germplasm, adapted in rainy and humid environments, represents a crucial reservoir of resistance genes against fungal diseases like head smut [22]. Still, it hasn’t been thoroughly explored to discover novel resistance genes against *S. reilianum*. This study evaluates the response of 153 Senegalese sorghum accessions to a newly identified pathotype 5 of *S. reilianum* through syringe needle inoculation (hypodermic injection). Furthermore, we investigated potential associations between head smut resistance and previously reported data in seed morphology and the seedling screening method established by Craig and Frederiksen [17,18,19,23]. To investigate these connections further, seed weight was measured for 59 lines, allowing us to assess any potential linkage between this trait and head smut susceptibility as well. To identify genes associated with resistance against P5 of *S. reilianum*, this study utilized GWAS using a dataset of over 297,876 genetic markers (SNPs) derived from genotyping-by-sequencing (GBS) available for these accessions.

## 2. Results

### Phenotypic Variation

Evaluating 153 Senegalese sorghum accessions against P5 of *S. reilianum*, we identified 63 lines completely free of disease (Table 1, Figure 1, Appendix A). The remaining accessions exhibited varying levels of susceptibility, with mean head smut incidence ranging from 1% to 25% (49 lines), 26% to 50% (26 lines), and exceeding 50% (15 lines). 

The X-axis displays the average disease incidence rate (%) calculated from disease evaluations. The Y-axis shows the number of sorghum lines within each score range. Appendix A provides further detailed information regarding sorghum responses against head smut P5.

## 3. Correlations between Head Smut and Seed Traits in Senegalese Accessions 

Pearson’s correlation analysis in Senegalese sorghum accessions revealed an intriguing link between head smut incidence rate (P5) and seed morphology traits [23], including newly measured seed weight (Table 2). Notably, larger seeds (area, perimeter, width, distance between IS, and CG) exhibited positive correlations with P5 incidence, suggesting greater susceptibility. Conversely, weak negative correlations were found with shape-related traits (length–width ratio and circularity). Additionally, head smut incidence showed a weak positive association with the seedling spot appearance rate. Further analysis of seed weight revealed moderate to strong positive correlations with size and width parameters (area, perimeter, length, width, and distance between IS and CG). However, seed weight negatively correlated with shape indices (length–width ratio and circularity) and brightness. Seed weight displayed a moderate positive correlation with head smut incidence.

Cluster analysis (Table 3) revealed an intriguing pattern: seed weight clustered with other seed size-related traits (cluster 1). Head smut incidence rate was grouped with two seed shape-related traits and the seedling head smut spot appearance rate (cluster 2) [18,19]. Interestingly, seedling head smut spot appearance time formed a distinct cluster (4) with seed brightness, while anthracnose resistance traits remained together in cluster 3.

### Genome-Wide Association Study

While no associations met the stringent Bonferroni threshold for genome-wide significance, six SNPs exhibited highly significant individual *p*-values (*p* < 1 × 10^−5^, −log_10_(*p*) *=* 5) and were considered potential candidates for head smut resistance markers (Figure 2, Appendix A). Appendix A provide a comprehensive list of these significant SNPs and their associated genes. By exploring the 100 kb genomic regions surrounding the six SNPs, we uncovered 35 promising new candidate genes potentially involved in sorghum’s defense against head smut. Some of these genes were associated with plant growth and development.

## 4. Discussion

Controlling head smut remains challenging due to factors like highly dormant soilborne teliospores and the emergence of diverse pathotypes [5,10]. Resistant cultivars offer the only effective control strategy [5,10]. However, new pathotypes necessitate continuous exploration of diverse germplasm for novel resistance sources [5,8,24]. In this study of 153 Senegalese sorghum accessions against P5 of *S. reilianum*, 63 disease-free lines were identified. The remaining 32% exhibited diverse susceptibility levels, from 1% to over 50% head smut incidence (Table 1 and Figure 1). This diverse response across the Senegalese collection highlights its potential as a valuable source of resistance against P5.

The correlation study showed a positive correlation between seed size and the P5 incidence rate, indicating increased susceptibility (Table 2). This contradicts our recent findings in a sorghum mini-core line study [25]. This discrepancy may stem from several possibilities, including using different sorghum accessions, distinct P5 pathotypes, or potential human error during manual inoculation. Further investigation is warranted to elucidate the underlying mechanisms driving this contrasting seed size–susceptibility relationship across various sorghum germplasm and head smut pathotypes. Conversely, weak negative correlations were found with shape-related traits (length–width ratio and circularity) (Table 2). Seed weight correlated positively with head smut incidence, supporting Boyles’ earlier observation [12] that heavier seeds tend to be more susceptible. This study further extends Boyles’ work [12] by demonstrating that the seed weight-head smut susceptibility link is also valid for P5, highlighting the potential importance of seed morphology in disease resistance (Table 2).

In cluster analysis, seed weight clustered with other seed size-related traits, as expected based on our previous work (Table 3) [23]. The head smut incidence rate is grouped with two seed shape-related traits and the seedling head smut spot appearance rate [18,19]. This suggests a potential link between seed shape and susceptibility to P5 head smut, warranting further investigation. Cluster 4 reveals a distinct association between seedling head smut spot appearance time and seed brightness, while Cluster 3 groups anthracnose resistance traits. This supports correlations between seemingly unrelated characteristics.

Through GWAS, six highly significant SNPs emerged as potential head smut resistance markers (Figure 2 and Appendix A). Further analysis revealed 35 promising candidate genes within the 100 kb genomic regions surrounding these SNPs, potentially contributing to sorghum’s defense against head smut (Appendix A). Of the 35 newly identified candidate genes, an F-box encoding gene on chromosome 2 stands out. Notably, F-box proteins regulate cell death and defense responses upon pathogen recognition in plants like tobacco and tomato [26]. Furthermore, F-box encoding genes have been consistently identified as top candidates in various sorghum GWAS studies against fungal pathogens, emphasizing their potential importance in disease resistance [27,28]. 

Another promising candidate from the identified genes encodes a zinc finger protein on chromosome 9. These proteins are ubiquitous plant regulators in both abiotic and biotic stress responses [29]. Studies in *Arabidopsis thaliana* and rice (*Oryza sativa*) have shown that defense proteins containing this domain are crucial for programmed cell death (PCD), a key mechanism for pathogen resistance [29]. Zhang et al. identified a specific C2H2-type zinc finger protein playing a key role in rice’s antioxidant defense system [30]. Likewise, Cui et al. discovered a C2H2 zinc finger gene family associated with cold and drought stress responses in sorghum [31].

*LOC8064738*, which encodes a Plasmodesmata Callose-Binding Protein 2 (PDCB2), represents an additional noteworthy candidate gene. Plasmodesmata (PD) are membrane-lined pores that connect adjacent cells to mediate symplastic communication in plants [32]. Studies suggest PD plays a role in the plant’s battle against pathogens [32]. Upon pathogen detection, plants can strategically close their PDs, hindering the spread of fungal and bacterial pathogens [33]. Notably, while viruses exploit PDs for movement, recent research suggests protein effectors from fungal pathogens can also utilize them [33]. 

Our GWAS analysis detected two candidate genes on chromosome 4 associated with the BZR1 protein. This protein is known to be part of the BZR1-EDS1 module, an essential component in plant growth with defense responses [34]. This module involves BZR1, a central transcription factor in brassinosteroid (BR) signaling, and EDS1, an essential positive regulator of plant innate immunity. Qi et al. observed that EDS1 interacts with BZR1 and suppresses its activity [34]. When EDS1 function is upregulated by a virulent *Pseudomonas syringae* strain or through salicylic acid treatment, it inhibits the expression of BR-responsive genes regulated by BZR1 and hinders BR-promoted growth [34].

An additional promising candidate is *LOC8085202*, encoding mevalonate kinase on chromosome 4. The mevalonate pathway is critical for cellular processes in both animals and plants, impacting growth, development, and stress response in plants [35]. In apple (Malus × domestica Borkh.), a recent study observed extensive transcriptome reprogramming upon inoculation, with 28.9% of expressed genes showing significant differential regulation, and the early activation of the mevalonate pathway correlates with reduced susceptibility, leading to smaller lesion diameters [36].

CUP-SHAPED COTYLEDON1 (*LOC8082811*) is associated with plant development. It activates the expression of LSH4 and LSH3, members of the ALOG gene family, specifically in shoot organ boundary cells [37].

*LOC8064189* encodes structural maintenance of chromosome protein 4 (SMC4). In plants, SMC proteins and their interacting proteins are crucial for various chromosomal functions, including sister chromatid cohesion, chromosome condensation, DNA repair, and recombination [38]. Research in Arabidopsis indicates that the SMC5/6 complex, a close relative of SMC4, serves functions beyond DNA repair [39]. This complex appears to play important roles in regulating plant development, responses to abiotic stress, suppression of autoimmune responses, and sexual reproduction [39].

Phytosulfokine, encoded by *LOC8085201* on chromosome 4, serves as a danger-associated molecular pattern recognized by the receptor PHYTOSULFOKINE RECEPTOR 1 (PSKR1) [40]. It initiates intercellular signaling to coordinate various physiological processes, particularly in the defense response to the necrotrophic fungus *Botrytis cinerea* in the tomato plant [40].

*LOC8084627* on chromosome 2 encodes the PIN-LIKES 7 (PIN7) gene. PIN proteins play critical roles in various developmental processes across land plants [41]. These processes range from the earliest stages of embryo formation (embryogenesis) to the shaping of organs (morphogenesis and organogenesis) and even extend to growth responses to environmental stimuli [41].

These genes, alongside the detailed candidate gene list in Appendix A, provide valuable resources for future research and breeding efforts to strengthen sorghum’s defense against head smut and other fungal diseases. The identified genes, many with established roles in plant defense, hold high potential for genuine involvement in fungal resistance. Several genes previously recognized as top candidates in other studies also emerged as candidate genes here. Not all candidate genes are associated with plant defense or stress responses, but the genes were associated with plant development and growth, potentially explaining the correlations between the head smut resistance and seed morphology listed in Table 2. Future research should explore these genes further in diverse sorghum germplasms against P5 but also against P6 and other variants to gain a deeper understanding. This broader investigation of sorghum’s defense against head smut will lead us to develop more resistant varieties.

## 5. Material and Methods

### 5.1. Sorghum Lines 

Overall, 153 Senegalese sorghum accessions (accession details in Appendix A) obtained from the USDA-ARS Plant Genetic Resources Conservation Unit in Griffin, Georgia, were assessed for their resistance to head smut P5. Both accessions and control lines (resistant BTx635 and susceptible BTx643) were screened using a mixture of two P5 isolates (#27 and #79). Planting occurred in 3-gallon pots filled with Metro Mix 200 (Sun Gro Horticulture, Agawam, MA, USA) and maintained under controlled greenhouse conditions with a 12 h photoperiod and a temperature of 25 ± 2 °C.

Sporidial inoculum preparation and inoculation followed the methods of Prom et al. [5]. Briefly, we cultured the pathotype 5 isolates in potato dextrose broth, shaking (150 rpm) for four days at 25 °C to promote yeast-like spore multiplication. Each isolate’s suspension was filtered through four layers of cheesecloth and combined in equal volumes, adjusting the concentration to 1 × 10^5^ spores/mL. Five plants per replicate were inoculated with 0.5–1.0 mL of the mixed suspension, injected below the apical meristem of 18–20-day old seedlings using a #22 G × 1” needle (Becton, Dickinson and Co., Franklin Lakes, NJ, USA) and a 30 mL syringe. To account for potential variability, the inoculation experiment was conducted twice with independent sets of plants.

### 5.2. Disease Evaluations

Disease assessment followed protocols established by Prom et al. [5,8]. Plants were evaluated at heading and categorized as resistant if they displayed fully developed grains without symptoms (no sori). Susceptible plants exhibited characteristic symptoms like smutted panicles, phylloidial miniature leaves with galls, “witches’ broom” panicles, or bleached sterile panicles. To confirm resistance, panicles from symptom-free plants were further examined by severing the main tiller and allowing side tillers to develop until flowering. Disease expression in side tillers indicated susceptibility, while continued symptom absence confirmed resistance. This ensured a binary (present or absent) disease rating for the host-pathogen system. Percent incidence was calculated as the number of infected plants per pot divided by the total inoculated plants multiplied by 100. Plant accessions above 50 percent infection were considered susceptible (S), and an infection rate less than 50 percent was considered resistant (R). During repeated times, if one trial is above 50 and the other is less than 50, it is considered various (V). For GWAS, R was converted into 0, S was converted into 1, and V was converted into 0.5. 

### 5.3. Statistical Analysis

We investigated potential associations between head smut resistance and eight previously characterized seed morphology traits (size, shape, color) identified in our earlier study in JMP Pro 15 (SAS Institute, Cary, NC, USA) [23]. Additionally, Senegalese accession data for anthracnose resistance at the fourth and eight leaf stages were analyzed to determine any correlations with the newly generated head smut data [27,42]. Seed weight (Appendix A) was phenotyped for 59 randomly selected lines to investigate further potential relationships by weighing five seeds each on a Mettler Toledo ML303E balance (20–22 replicates per line). Pearson’s correlation analysis was conducted to identify pairwise correlations between seed morphology and anthracnose resistance traits and P5 head smut incidence rate. Finally, all phenotypic data were used for a clustering analysis to explore the relationships between all traits.

### 5.4. Genome-Wide Association Study

The SNP data was extracted from a sorghum SNP dataset, originally genotyped using genotyping-by-sequencing (GBS), based on the sorghum reference genome version 3.1.1 [43,44,45]. Missing data were imputed using Beagle 4.1 [46] and further filtered for minor allele frequency of at least 0.01. 

Genome-wide association analysis was conducted using the R-package, Genome Association and Prediction Integrated Tool (GAPIT) version 3 [47]. We employed the multi-locus mixed model (MLMM) [48] with population stratification controlled by principal component analysis (PCA). The optimal number of principal components was determined through a Bayesian information criterion-based analysis within GAPIT. For the association analysis in GAPIT, the following parameters were used: PCA.total = 3 and model = MLMM, with other parameters at their default values. Significant SNP-trait associations were identified with a threshold of −log10 (*p*-value) of 5 or greater. Pairwise LD (r^2^) between the significant SNPs and nearby SNPs (100 kb upstream and downstream) was estimated using PLINK1.9 [49]. LD blocks were defined by merging SNPs with an r^2^ value of at least 0.4 and visualized using Haploview 4.2 [50]. LD blocks represent genomic regions where SNPs are co-inherited due to strong LD. If the LD block was smaller than 100 kb, the genomic region was extended up to 100 kb for candidate gene identification based on the sorghum reference genome annotation from the Phytozome 12 [https://phytozome.jgi.doe.gov (accessed on 15 January 2024)] (version 3.1.1, GCF_000003195.3) [51]. 

## 6. Conclusions

By analyzing 153 Senegalese sorghum accessions, we uncovered novel links between seed morphology and the newly identified P5 strains of *S. reilianum*, with stronger susceptibility observed in larger seeds. This finding contrasts with our study in sorghum mini core lines [25] but highlights potential germplasm-specific variation in seed size–susceptibility relationships. Notably, systemic syringe inoculation exhibited only a weak linkage to seedling spot appearance, suggesting diverse defense mechanisms throughout sorghum growth, as outlined by Craig and Frederickson [17]. Moreover, GWAS analysis identified SNP markers and candidate genes potentially associated with the P5 incidence rate, suggesting opportunities for future functional validation studies. Overall, this work unveils intriguing connections between seed morphology, defense mechanisms, and head smut resistance, warranting further exploration to enhance disease resistance.

## Figures and Tables

**Figure 1 plants-13-00977-f001:**
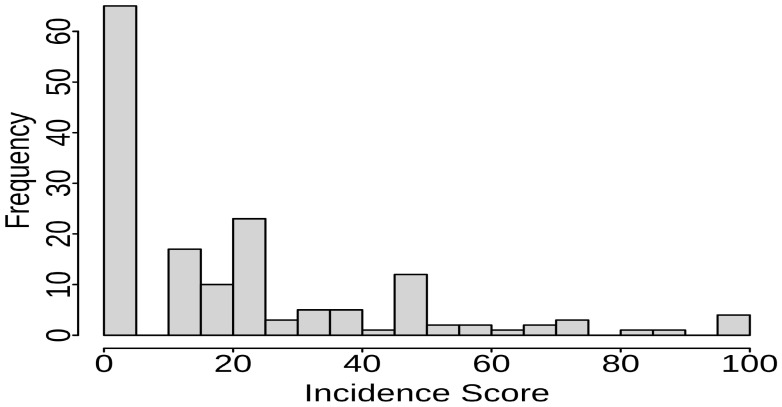
Distribution of disease incidence in Senegalese sorghum accessions.

**Figure 2 plants-13-00977-f002:**
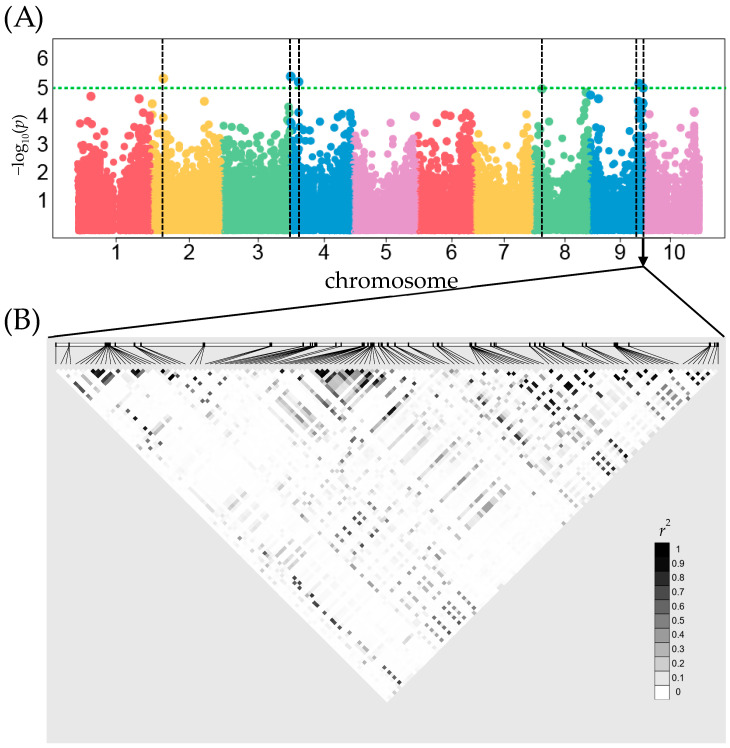
Manhattan plot and linkage disequilibrium (LD) plot for head smut susceptibility. (**A**) Manhattan plot showing significant SNPs associated with head smut susceptibility across the genome. Dots represent SNP markers. The green horizontal dashed line indicates the genome-wide significance threshold (*p* < 1 × 10^−5^, −log_10_(*p*) *=* 5). Black vertical dashed lines show the positions of quantitative trait nucleotides (QTNs). (**B**) LD plot was constructed using Haploview 4.2 for the 100 kb genomic region surrounding the significant SNPs. The color scale indicates LD strength expressed as r^2^, with black representing perfect LD (r^2^ = 1) and white representing no LD (r^2^ = 0). Appendix A provide detailed information about the candidate SNPs and their associated genes.

**Table 1 plants-13-00977-t001:** Highly resistant sorghum accessions to head smut pathotype 5. Among 153 sorghum accessions evaluated for resistance to head smut pathotype 5 of *S. reilianum,* 63 exhibited complete resistance.

Sorghum Accessions Resistant to P5 Head Smut (*S. reilianum*)		
PI 514297	PI 514376	PI 514433
PI 514310	PI 514377	PI 514434
PI 514317	PI 514379	PI 514436
PI 514320	PI 514381	PI 514437
PI 514323	PI 514382	PI 514438
PI 514338	PI 514388	PI 514439
PI 514340	PI 514392	PI 514449
PI 514343	PI 514394	PI 514452
PI 514345	PI 514397	PI 514455
PI 514346	PI 514398	PI 514460
PI 514349	PI 514399	PI 514461
PI 514351	PI 514401	PI 514463
PI 514353	PI 514403	PI 514465
PI 514354	PI 514404	PI 514466
PI 514355	PI 514411	PI 514467
PI 514362	PI 514412	PI 514468
PI 514364	PI 514414	PI 514471
PI 514366	PI 514418	PI 514473
PI 514368	PI 514419	PI 514474
PI 514373	PI 514424	PI 514475
PI 514375	PI 514428	PI 609251

**Table 2 plants-13-00977-t002:** Correlations between head smut incidence rate (P5), seed morphology traits, seed weight, and seedling responses to head smut (P5). Pearson’s correlation coefficients are shown between mean head smut incidence rate, seed weight, previously identified seed morphology traits, and seedling responses to head smut (P5). Significance levels are denoted by asterisks: *** = *p* < 0.0001, ** = *p* < 0.001 and * = *p* < 0.05.

	Head Smut Mean Incidence	Seed-Weight
Head smut mean incidence	1.00 ***	0.49 **
Seed-Area size	0.16 *	0.61 ***
Seed-Perimeter length	0.25 **	0.67 ***
Seed-Length	0.13	0.40 **
Seed-Width	0.22 **	0.74 ***
Seed-Length–width ratio (LWR)	−0.23 **	−0.72 ***
Seed-Circularity	−0.27 **	−0.30 *
Seed-Distance between the intersection of length and width (IS) and center of gravity (CG)	0.32 ***	0.53 ***
Seed-Brightness	−0.04	−0.41 **
4-leaf anthracnose average score	−0.06	0.63
4-leaf anthracnose highest score	0.01	0.61
8-leaf anthracnose average score	0.02	0.48
Seedling head smut spot appearance rate	0.22 *	0.16
Seedling head smut spot appearance time	−0.03	−0.17
Seed-Weight	0.49 **	1.00 ***

**Table 3 plants-13-00977-t003:** Clustering of seed morphology, anthracnose resistance, and head smut traits in sorghum accessions. Cluster analysis based on the newly investigated traits (head smut incidence rate and seed weight), and previously known seed morphology and anthracnose resistance traits identified four distinct clusters. Bolded entries indicate newly acquired phenotypes in this study.

Cluster	Members	R^2^ with Its Own Cluster	R^2^ with the Next Closest Cluster
1	Seed-Width	0.99	0.03
	Seed-Area size	0.94	0.02
	Seed-Perimeter length	0.94	0.06
	Seed-Length	0.72	0.01
	Seed-LWR	0.62	0.03
	Seed-Weight	0.58	0.26
2	Seed-Circularity	0.64	0.02
	Seedling head smut spot appearance rate	0.50	0.06
	Seed-Distance between IS and CG	0.55	0.21
	Head smut mean incidence	0.35	0.07
3	4-leaf anthracnose highest score	0.91	0.01
	4-leaf anthracnose average score	0.92	0.04
	8-leaf anthracnose average score	0.04	0.01
4	Seedling head smut spot appearance time	0.53	0.01
	Brightness	0.53	0.03

## Data Availability

The raw data are contained within Appendix A.

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
