# Peer review of "Genome-Wide Association Analysis Uncovers Genes Associated with Resistance to Head Smut Pathotype 5 in Senegalese Sorghum Accessions"

_plants, 2024, doi:10.3390/plants13070977_

Round 1

Reviewer 1 Report

Comments and Suggestions for Authors

The manuscript gives new information about resistance to head smut in sorghum. I think it requires “minor revisions” for publication.

1 Phenotypic variation test

Abstract (L. 15-16), L. 96-100

Please reconsider and correct the manuscript.

2 Table 2

    I think Table 2 is unnecessary.

3 L.115-116

    Are the correlations weak between head smut mean incidence and shape related traits?

4 L.145-147, discussion

    I think authors should discuss the genes related with seed shape and genes including within “100 kb genomic regions surrounding the six SNPs.

Author Response

Thank you for your valuable feedback. We have carefully revised the manuscript to reflect your suggestions. 

Reviewer 2 Report

Comments and Suggestions for Authors

The Figs/supplementary data should be cited appropriately while discussing the specific genes/loci in the Discussion Section.

Can the 6 significant SNPs be compiled in a separate table for reference? The authors should provide a clear rationale of discussing only these 6 genes from Table S2.

Please update the information in the figure legends for the genes discussed in the text. This will be helpful for the readers.

Can you provide a viable scientific explanation for the current results contrasting the authors previous results (ref 25)? Do you think use of different sorghum accessions, P5 pathotypes should produce contrasting results. 

Author Response

(The authors gave the same response as above.)
